# Optimizing Nitrogen Application for Jojoba under Intensive Cultivation

**DOI:** 10.3390/plants12173132

**Published:** 2023-08-31

**Authors:** Arnon Dag, Shamir Badichi, Alon Ben-Gal, Aviad Perry, Noemi Tel-Zur, Yonatan Ron, Zipora Tietel, Uri Yermiyahu

**Affiliations:** 1Gilat Research Center, Agricultural Research Organization, Volcani Institute, Rishon LeTsiyon 8528000, Israel; badichi@post.bgu.ac.il (S.B.); bengal@agri.gov.il (A.B.-G.); aviadper@post.bgu.ac.il (A.P.); yron@volcani.agri.gov.il (Y.R.); tietel@volcani.agri.gov.il (Z.T.); uri4@volcani.agri.gov.il (U.Y.); 2The R.H. Smith Institute of Plant Science and Genetics in Agriculture, Faculty of Agriculture, Food and Environment, The Hebrew University of Jerusalem, Rehovot 7610000, Israel; 3The Albert Katz International School for Desert Studies, The Jacob Blaustein Institutes for Desert Research, Sede Boqer Campus, Ben-Gurion University of the Negev, Sede Boker 8499000, Israel; 4French Associates Institute for Agriculture and Biotechnology of Dryland, The Jacob Blaustein Institutes for Desert Research, Ben-Gurion University of the Negev, Sede Boqer Campus, Sede Boker 8499000, Israel; telzur@bgu.ac.il

**Keywords:** diagnostic leaf, fertilization, plant nutrition, productivity, *Simmondsia chinensis*, vegetative growth

## Abstract

Although jojoba (*Simmondsia chinensis*) has been cultivated for years, information on its N requirements is limited. A 6-year study of mature jojoba plants grown under field conditions with an intensive management regime evaluated the effect of N application rate on plant nutrient status, growth, and productivity, and nitrate accumulation in the soil. Five levels of N application were tested: 50, 150, 250, 370, and 500 kg N ha^−1^. Fertilizers were provided throughout the growing season via a subsurface drip irrigation system. Leaf N concentration, in both spring and summer, reflected the level of N applied. A diagnostic leaf (youngest leaf that has reached full size) concentration of 1.3% N was identified as the threshold for N deficiency. Increasing rates of N application resulted in higher P levels in young leaves. Plant K status, as reflected in the leaf analysis, was not affected by N treatment but was strongly affected by fruit load. Vegetative growth was inhibited when only 50 kg N ha^−1^ was applied. Soil analysis at the end of the fertilization season showed substantial accumulation of nitrate for the two highest application rates. Considering productivity, N costs, and environmental risk, 150 kg N ha^−1^ is the recommended dosage for intensively grown jojoba. N deficiencies can be identified using leaf analysis, and excess N can be detected via soil sampling toward the end of the growing season. These results and tools will facilitate precise N fertilization in intensive jojoba plantations.

## 1. Introduction

Jojoba (*Simmondsia chinensis*) is a perennial, dioecious, evergreen shrub suitable for cultivation in arid and semi-arid areas. It is native to the arid regions of the Sonora and Mojave deserts in the southwestern United States and northwest Mexico. In its natural habitat, jojoba usually grows in coarse desert soils on the rocky slopes of mountains in areas with 100–300 mm annual rainfall [1]. The transcriptomic, proteomic, and metabolomic pathways that confer jojoba’s tolerance to the extreme heat-stress conditions in its habitat have been recently characterized [2].

Jojoba oil composition was first analyzed in 1933, revealing its potential as a replacement for sperm whale oil [3]. Jojoba is the only plant species known to accumulate wax esters in seeds instead of triglycerides [4]. Its commercial potential comes from the odorless and colorless wax extracted from its seeds and used in a variety of products, including cosmetics, pharmaceuticals, lubricants, and petrochemicals [5,6,7].

Nitrogen (N) is a major nutritional element in plants, and is required in many physiological and developmental processes, such as photosynthesis [8], metabolite biosynthesis [9], and flowering [10]. Most N uptake in perennial crops occurs during the intensive vegetative growth and postharvest stages [11], in March–May and September–December for jojoba growing in the northern hemisphere [12]. On the other hand, N pollution has been identified as one of the top emerging threats faced by humanity and the planet due to its impact on climate, the environment, and public health [13,14], and N excess originating from agricultural systems is considered to be a major source of this pollution [15].

Despite the increasing importance of jojoba cultivation and its transformation from an extensive, rain-fed crop to an intensive, irrigated one, few studies have focused on its N requirements. Jojoba has relatively low reliance on fertilizers and has traditionally been produced in soils with marginal fertility compared to other crops [16,17]. Early work in Arizona and California showed a negligible response of jojoba to N fertilization in terms of seed yield [18] and a relatively modest amount of 58 kg N ha^−1^ year^−1^ was suggested to be sufficient for a yield of 3000 kg seeds ha^−1^ [19]. Nonetheless, more recent studies have demonstrated a positive effect of N on commercial jojoba plantations. N, phosphorus (P), and potassium (K) fertilization in jojoba (80, 8, and 80 kg ha^−1^, respectively) shortened the period to fruit production and provided better vegetative growth compared to unfertilized plants [20]. However, this experiment was conducted with old cultivars that had limited productivity and were irrigated only once a month. This response was attributed to the combined effect of fertilization and irrigation. Benzioni (1995) illustrated the jojoba plant’s response to nitrate fertilizer by measuring nitrate reductase activity and found much higher levels after application, mainly during growth and fruit filling [21]. More recent research conducted under North Sinai conditions evaluated the effect of N foliar spray on yield and seed quality in jojoba. The treatments included two rates of a N-P-boron (B) foliar spray (1%-0.75%-0.4% and 1.5%-1.25%-0.8% of N, P, and B, respectively) applied three times (October, March, and April) and a control treatment with no fertilizer application. Higher rates of N (together with P and B) were found to increase growth, yield, and seed oil content [22]. Research conducted in Arizona on 2-year-old local cultivars treated with N as a solid fertilizer showed a correlation between N dosage and seed yield. The levels were 0, 30, and 60 N kg ha^−1^ year^−1^ for the first 4 years, and increased to 0, 60, and 120 N kg ha^−1^ year^−1^ for the remaining 6 years of the study. Differences in plant height between the treatments were only significant in one year, whereas seed yield increased with increasing N application in 4 of the 7 years. An effect on leaf N concentration was visible after 3 years of N application [23].

As mentioned earlier, Optimizing N application is essential for maximizing productivity, on the one hand, and reducing environmental contamination, on the other. Hence, the objective of the current study was to determine the N fertilization requirements of modern intensively cultivated jojoba by evaluating the effect of N availability on vegetative and reproductive development. In contrast to previous studies, in which irrigation was via furrow and N was applied once or twice annually [23], or where N treatments were combined with differential P, B, and K levels [20,22,24,25]—making it difficult to isolate the effects of N—this study focused on applying N throughout six seasons via drip irrigation (fertigation).

## 2. Results

### 2.1. Leaf Analysis

Differential N fertilization treatments were initiated in June 2016. At the first leaf sampling 2 months later (8 August 2016), leaf N concentration was similar for all treatments (average of 1.8%). Eight months later, at the end of March 2017, N accumulation in the leaves started to correspond to the N level applied, with the N50 treatment showing significantly lower leaf N compared to treatments N370 and N500. This trend continued in the next year’s spring and summer analysis but with no significant differences between treatments above N50 (Figure 1). As the experiment progressed, the differences increased moderately, and mean leaf N concentration, in both spring and summer, reflected the level of N applied.

P concentration changed with leaf age in both spring and summer (Figure 2). The concentrations of P in old and young leaves sampled in the summer were similar between treatments, but with a significantly higher level in 2020 compared to 2019. Young leaves sampled in the spring of both years responded to the N level applied, with a higher rate of N application resulting in higher P levels.

Leaf K level was similar in all treatments and was not affected by N availability; however, it was strongly influenced by year of sampling. Figure 3 shows the year-to-year fluctuations in K level in old leaves sampled in the summer for treatment N250 with the associated yield.

The effects of the N fertilization treatments on the concentration of micronutrients in old (O) and young (Y) leaves are presented in Appendix A. The concentrations in 2016 were similar between all treatments. A significant positive response was measured for Ca (2018 Y, 2020 Y), S (2020 O, 2020 Y), Mn (2018 O, 2018 Y, 2020 Y), and Mo (2020 O, 2020 Y).

### 2.2. Soil Nitrogen

Nitrate accumulation was detected in the soil in the N370 and N500 treatments, with significantly higher levels in the former (Figure 4) (*p* < 0.05). The soil was sampled in September 2020, after the annual amounts of fertilizer had been applied to all treatments. The fact that the N500 treatment’s annual fertilizer application was completed some 3 weeks prior to that of the N370 treatment and sampling might explain the differences. Insignificant amounts of N were measured in the other treatments. Higher N was detected in the deeper soil layers. Ammonium levels were much lower (10-fold on average) and are therefore not presented.

### 2.3. Vegetative Growth

Vegetative growth, as determined by branch elongation, for the N50 treatment was insignificantly lower in 2020 than for the other treatments (Figure 5A). In 2021, the differences between treatments were more pronounced and were strongly linked to the applied N level (Figure 5B).

N fertilization treatments affected the weight of pruned branches in 2018, 2020, and 2021, with statistically significant differences observed between treatments N50 and N370. No effect was visible in 2019, except that the mean value in treatment N50 was ~50% lower than in the other treatments (Figure 6). The highest accumulated weight during 2018–2021 was found for treatment N370 (11.13 kg plant^−1^), which was not significantly different from any of the other treatments, except N50.

### 2.4. Productivity

Until 2019, no differences were observed in seed weight as a function of increasing N fertilization. Seed weight in treatment N50 exhibited a significant decrease in 2019, 2020, and 2021 compared to that in treatment N370 (Figure 7). In addition, seed weight in 2020 revealed a response curve that matched the level of N applied. In ‘On’ years (years with a relatively high yield: 2017, 2019, and 2021), seed weight was significantly reduced in all treatments. The N fertilization rate did not influence the seed wax content. However, a statistically significant difference was present between seasons, with 2017 having the highest value, averaging 53.7%, and 2018 having the lowest, at 48.3%; the overall average percentage of wax in the seed for all treatments from 2016 to 2020 was 50.84.

N treatments also did not affect fruit set. The total mean fruit sets for all treatments were 74.8, 76.5, 78.7, 86.2, and 71.5% in 2017, 2018, 2019, 2020, and 2021, respectively.

The response of annual jojoba seed yield to increased N fertilization was limited, with statistically significant differences occurring in ‘Off’ years (low-yield seasons: 2018 and 2020) between treatments N50 and N370 (Figure 8), and in 2021 between treatments N50 and N250. The effect of alternate bearing is visible, with an average yield increase of 2.9 times for all treatments in ‘On’ compared to ‘Off’ years. The terms ‘Off’ and ‘On’ years are used for crops with typical biennial behavior where a low-yield season (‘Off’ year) is followed by high-yield season (‘On’ year). The 5-year accumulated yield curve showed a polynomial response to increasing N application (Figure 9), peaking at around 300 kg N ha^−1^. However, the yield increase between the N150 and N370 treatments, representing more than double the N application level, was only 6.5%.

## 3. Discussion

This work brings some of the first results from an evaluation of N demand in jojoba crops, and possibly the very first results for jojoba grown under modern intensive, irrigated conditions. The results and the study’s conclusions are expected to be important to the rapidly growing and intensifying jojoba industry.

Five years of biannual monitoring of leaf mineral levels, from the summer of 2016 to the spring of 2021 (Figure 1, Figure 2 and Figure 3), revealed higher leaf N concentrations corresponding to increasing N fertilization levels. These results indicate that the treatments affected N availability and plant nutritional status. Previous work conducted on N nutrition in olives [26] and pomegranate [27] showed only limited N accumulation in the leaves following high levels of N fertilization. This suggests that while leaf N analysis can successfully identify crop N shortages, it is less useful for the identification of surplus crop N. In our study, N levels were indeed well correlated with N treatment at lower levels; the extremely high level of N500 did not differ consistently from those in the N37O treatment, similar to the aforementioned studies with olive and pomegranate.

In their previous studies on jojoba N fertilization, both [24] and [23] reported higher leaf N concentrations than the 5-year average observed in our N500 treatment (1.66 and 1.69% in the spring and summer sampling, respectively), despite their annual application rate being only 50 and 60 kg N ha^−1^, respectively. The authors of [24] reported levels ranging from 0.37 to 3.06% N in diagnostic leaves in spring and 0.24 to 2.11% in summer. A possible explanation for our relatively lower N levels in response to higher N application may lie in the age of the plants. Both previous studies were conducted on much younger plants than ours, at 2 years from seeding in [24], and in 2-year-old plants in [23] at the beginning of their study; here, the plants were fully mature—14 years old at the beginning of the study. The relatively high N in the leaves in the other two previous studies may be explained by the fact that young plants tend to have higher N levels due to their intensive vegetative growth [28].

Upon comparing the spring to the summer leaf sampling, there seems to be large differences between years in spring, which might be explained by differences in when the spring growth begins [29]. Hence, summer (July; Figure 1B), when the levels between years are more balanced, might be a better time to sample diagnostic leaves to assess jojoba plant N status, as also proposed by [23,30]. We further suggest 1.3% N as the threshold for N deficiency in jojoba because this value distinguished between the N50 treatment, which had relatively low productivity, and the other treatments, which all showed full production with apparently sufficient N levels.

N availability affected plant P status, as reflected by the analysis of young leaves in the spring (Figure 2). The factors involved in increased plant P with increasing N availability include increased solubility of soil P caused by a pH decrease in the root zone (due to increasing ammonium concentration) and increased root growth, leading to an increased ability to take up P [31]. In the current experiment, P was applied annually at a constant rate of 100 kg ha^−1^ in all treatments, so that it would not be a limiting factor for plant performance. Therefore, we found that higher N availability increases P uptake, especially at the beginning of the season. This phenomenon was reported in *Larix gmelinii* while N was artificially added [32], and in leguminous trees following atmospheric N fixation [33], but this is the first time it has been reported in jojoba.

A strong impact of fruit load on K status in leaves (Figure 3) has been previously reported for jojoba [12], as well as for other crops, including olive [34], which is explained by developing fruit being a strong K sink. Our results indicate that a large amount of K is removed in ‘On’ years with the harvest of jojoba seeds. ‘On’ year seed yield reached an average of 7 ton ha^−1^ with 0.37% K, reaching an average of 25.9 kg K ha^−1^. The large fluctuation in K between ‘Off’ and ‘On’ years (Figure 3) suggests a need to consider fruit load when setting K standards for diagnostic leaves.

The N fertilization treatments affected plant macro- and micronutrient status, as shown in Appendix A. Leaf S, Ca, and Mn increased, while Na, Cu, and Mo decreased as a function of a higher N application rate. Higher N increases root and canopy growth, which affects both mineral uptake and the ability to meet nutritional demands. Except for Mo, the concentrations of all micro- and macroelements were strongly influenced by the age of the sampled leaves.

Nitrate is known to be highly mobile in the soil and to accumulate in the deeper soil layers, as measured in the present experiment (Figure 4). Such N transport and accumulation was apparent only for the N370 and N500 treatments, because almost no nitrate was detected in the soil after the fertilization season in the other treatments. Residual N indicates the application of excessive amounts that the plants do not take up and utilize. Aside from economic waste, N fertilization costs are substantial in horticultural crops, and nitrate leaching and the contamination of deep soils and groundwater are of major global concern [35,36]. The higher accumulation of nitrate in the N370 treatment compared to the N500 treatment can likely be explained by different fertilization timing between treatments, resulting in no fertilizer application 3 weeks before sampling in the N500 treatment.

Both vegetative growth parameters—branch elongation (Figure 5) and pruning material weight (Figure 6)—clearly showed that N50 was a deficient treatment that inhibited vegetative growth. The importance of N availability for vegetative growth has been well documented [37]. This inhibited growth was reflected later in lower productivity, credited to flowers being produced on the previous year’s growth [29]. Branch elongation better reflected the order of the treatments compared to pruning material weight. Branch elongation was more responsive to treatments in the ‘On’ year of 2021 (Figure 5B), than in the ‘Off’ year of 2020 (Figure 5A). It seems that under high fruit load, when sinks are strong, N availability has a greater effect on growth than under low fruit yield, when there is less competition between vegetative and reproductive growth. Thus, the effect of N treatment was relatively small in ‘Off’ years. A similar phenomenon was found in olives, where irrigation levels affected oil productivity in ‘On’ but not ‘Off’ trees [38]. The high pruning material weight obtained in 2020 (Figure 6) was probably a result of accelerated growth in that year due to reduced fruit load. Jojoba fruits compete with vegetative growth for carbohydrates and nutrients [12], and therefore, in ‘Off’ years, reduced competition allows for better growth.

With respect to fruit load, we found typical biennial bearing behavior in the current study for the jojoba plant, where 2016, 2018, and 2020 were ‘Off’ seasons and 2017, 2019, and 2021 were ‘On’ seasons (Figure 8). This phenomenon has been described for jojoba in previous studies [39,40]. The main explanation for biennial bearing in jojoba is that the vegetative growth (which carries the reproductive nodes for the next year’s flowers) is affected by yield load: under low yields, the plant produces many more new nodes due to the accelerated vegetative growth [29]. The effect of crop load on vegetative development was particularly visible in the pruning material weight in the current study (Figure 6). These facts might also explain the finding that the effect of low N on productivity is more pronounced in the ‘Off’ year, which is the outcome of the inhabited growth in the previous ‘On’ year.

We found no effect of N application rate on wax content in the seeds. In contrast, [23] found that increased N causes a slight but significant reduction in seed wax content. Those authors further reported an increase in seed weight with increasing N in some of the years, a phenomenon that we also observed in the last 3 years of the current study, in 2019–2021 (Figure 7).

We identified 150 kg N ha^−1^ as the optimal application rate with respect to fertilizer costs, crop productivity, and risk of environmental contamination with excess nitrate [38]. This value is higher than that documented for jojoba in [23], which referred to 120 kg N ha^−1^ as a high application rate in mature plantations. The higher productivity in the current study (ca. 5.0 kg seed/plant) compared to that in the study by Nelson and Watson (ca. 1.5 kg seed/plant) is probably associated with higher N demand. It is interesting to note that an application rate of 150 kg N ha^−1^ was also found to be optimal in olive, another perennial oil crop, under intense production with fertigation under field conditions [26]. Other crops, like pomegranates and apple, reported lower N fertilization requirements [41,42].

## 4. Materials and Methods

### 4.1. Experimental Design

The experiment was carried out in a 14-year-old commercial jojoba plantation (cv. Hatzerim) owned by Jojoba Israel near Beer Sheva (31°14′45.9″ N 34°43′18.2″ E). Plant density was 1100 plants ha^−1^ with 4.5 m between rows and 2 m between plants in a row. The soil texture was loam, consisting of 36% sand, 18% clay, and 46% silt. The soil’s water saturation percentage and CaCO_3_ content were 37.3% and 19%, respectively. Plants were mechanically pruned yearly via hedging after harvesting, around November, to enable machinery access and light penetration into the canopy [43].

An experimental orchard was designed to test different fertilization treatments in a randomized-block statistical scheme, with five plots per block (a total of 25 plots), producing five plots per treatment (replications). Each plot was composed of three rows with nine plants in a row. Only the five middle plants in the middle row of each plot were measured. The rest of the plot was considered margins to reduce edge effects.

Separate irrigation and fertilization controllers and pumps were installed for the experiment at the beginning of 2016. Each fertilization treatment had its own pump, water meter, and liquid-fertilizer container. Fertilizer solutions were prepared according to the treatment specifications and supplied by local fertilizer companies Israel Chemical Ltd. and Deshen Gat. Irrigation was delivered weekly from November to March and twice a week during the rest of the year. The irrigation was with fresh water (EC = 0.30 dS m^−1^) via a subsurface drip irrigation system (Netafim, Hatzerim, Israel), with two laterals buried at a 40 cm depth in every other row. The applied irrigation volumes were based on returning reference evapotranspiration multiplied by a crop coefficient of 0.5. Daily meteorological values were obtained from the Israel Meteorological Service’s (ims.gov.il; accessed on 1 February 2023) Gilat meteorological station, located 11 km from the experimental field. Average irrigation amounts for all treatments were 572, 620, 713, 737, 693, and 537 mm annually in 2016, 2017, 2018, 2019, 2020, and 2021, respectively.

N fertilization was applied continuously and proportionally between March and November via the irrigation system (fertigation), with respect to the expected irrigation amount, determined based on the 10-year daily average evapotranspiration from the Israel Meteorological Service data. Treatments included five N application rates (50, 150, 250, 370, and 500 kg h^−1^) (Table 1). The ratio between ammonium and nitrate forms (N-NH_4_ and N-NO_3_, respectively) in the fertilization solution was about 1:1 in all treatments. For all treatments, P and K were set at 100 and 300 kg ha^−1^ year ^−1^, respectively. Iron (Fe), manganese (Mn), zinc (Zn), copper (Cu), and molybdenum (Mo) were given with the fertilizer as chelate (3%), with concentrations of 0.3, 0.15, 0.07, 0.01, and 0.008 g L^−1^, respectively. Fertilization treatments were initiated on 22 June 2016, after 6 months of intentional N depletion with no fertilizer application.

### 4.2. Measurements

Measurements were taken annually to evaluate the nutritional status and vegetative and reproductive development of the measured jojoba plants. In addition, soil N was sampled and analyzed toward the end of the experiment.

#### 4.2.1. Leaf Analysis

Leaves were sampled in March (spring) and July (summer). Sampling included all measured plants in the plot from both sides of the row, choosing the youngest leaves that had reached full expansion (diagnostic leaves). In the years 2019 and 2020, there was additional sampling of the youngest expanded leaves prior to the development of waxy cuticles. The young leaves were analyzed for P, calcium (Ca), magnesium (Mg), sodium (Na), chloride (Cl), sulfur (S), B, Cu Fe, Mn, Zn, and Mo. Leaf samples were rinsed with deionized water, dried at 70 °C for 3 days, and ground to a homogeneous powder. A 0.1-g sample was digested with sulfuric acid (1.2 mL) and peroxide (0.5 mL). Concentrations of N and P were determined using an automated discrete chemical analyzer (Gallery Plus, Thermo Scientific, Vaata, Finland). K concentration was determined via atomic absorption spectrophotometry (Perkin-Elmer 460). In addition, leaf element concentrations were determined for the summer sampling after digesting the ground material with nitric acid and hydrogen peroxide and analyzing it using a 5100 ICP-OES instrument (Agilent Technologies, Mulgrave, Australia).

#### 4.2.2. Soil N Analysis

Soil samples were collected from four depths (0–20, 20–40, 40–60, and 60–80 cm) using a manual auger from all experimental fertilization plots to assess soil N after the 2020 fertilization amounts were given and before the 2021 fertilization period began. Saturated paste soil solution extract was used to determine nitrate concentration using the automated discrete chemical analyzer (Gallery Plus).

#### 4.2.3. Vegetative Growth

In 2020 and 2021, branches with young and fully developed leaves were randomly chosen in each plot. Ten branches per plot, five from each side (east and west), were marked with tape behind the second node from the branch tip. Marking took place in February when new shoots were starting to emerge. The increase in branch length was measured once a month as the length from the first node to the branch tip.

Pruned material was weighed annually toward the end of November after commercial, mechanized pruning had been uniformly conducted in all experimental plots in the 2018–2020 seasons. Branches were collected from the ground and weighed to evaluate and quantify the seasonal vegetative growth per plot.

#### 4.2.4. Seed Weight and Wax-Content Analysis

A random sample of 100 seeds was taken from the harvested yield of each plot and weighed to determine the average seed mass. The seeds were then dried in an oven at 70 °C for 3 days. The dried samples were ground, weighed and transferred to a paper tube capped with a ball of cotton wool to determine the wax content using Soxhlet extractor apparatus heated to 70 °C and filled with n-hexane for 6 h. The tube was oven-dried and weighed again, with the weight difference after the wax extraction representing the wax content.

#### 4.2.5. Fruit Set and Yield

To determine fruit set, branches from each plot were marked before flowering (December–January) and the number of flower buds on each branch was recorded. In May, after pollination and fruit set, the branches were further surveyed and the number of fruits per branch was counted. Nine, four, and ten branches per plot were marked for the seasons 2017–2018, 2019, and 2020–2021, respectively.

Harvesting was carried out after the seeds had fully matured. The jojoba plants were shaken using a commercial mechanical canopy shaker to release the attached fruit. Dropped fruits from each plot were collected in boxes, and then, separated from the leaves and litter using a conveyor belt and a blower. The yield per plant was calculated by dividing the yield per plot by the number of measured plants in each.

### 4.3. Data Analysis

JMP statistical software (JMP Pro version 15.0.0) was used for statistical analysis of the data. Five plots per treatment represented repetitions for the statistical analysis. Significance was set at *p* < 0.05, and treatments were compared using the Tukey-HSD test.

## 5. Conclusions

The provision of N via fertigation throughout the critical season for reproductive and vegetative growth was found to be highly efficient with regard to its uptake and utilization by jojoba. The N treatments were reflected in the leaf analyses, with clear recognition of deficient levels, whereas excess levels were only slightly revealed. Hence, precise N fertilization should be based on the following dual-monitoring approach: the assessment of N concentration in diagnostic leaves to detect deficiencies, and the assessment of soil N at the end of the fertilizer-application season to identify surpluses. Integrating productivity data with the results of accumulated nitrate in the soil implies that 150 kg N ha^−1^ is an optimal annual application rate for jojoba under intensive irrigated cultivation regimes.

## Figures and Tables

**Figure 1 plants-12-03132-f001:**
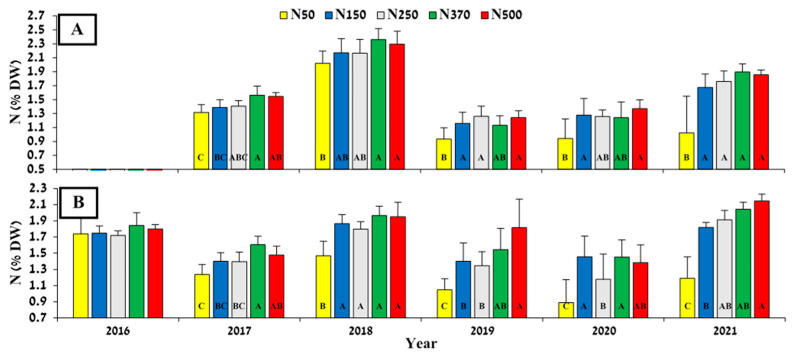
Effect of N-fertilization rate on leaf N concentration (% dry weight) measured in spring (**A**) and summer (**B**) of each experimental year. Values are mean ± SE of five plots per treatment. Different letters indicate significant difference (*p* < 0.05, Tukey-HSD) between treatments for each sampling date separately.

**Figure 2 plants-12-03132-f002:**
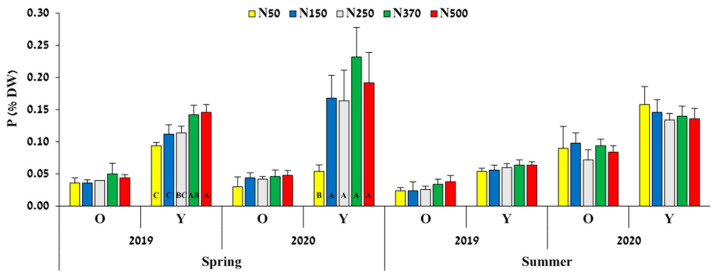
Effect of N-fertilization rate on P concentration (% dry weight) in old (O) and young (Y) leaves measured in the spring and summers of 2019 and 2020. Values are mean + SE of five plots per treatment. Different letters indicate significant difference (*p* < 0.05, Tukey-HSD) between treatments for each age and sampling date separately.

**Figure 3 plants-12-03132-f003:**
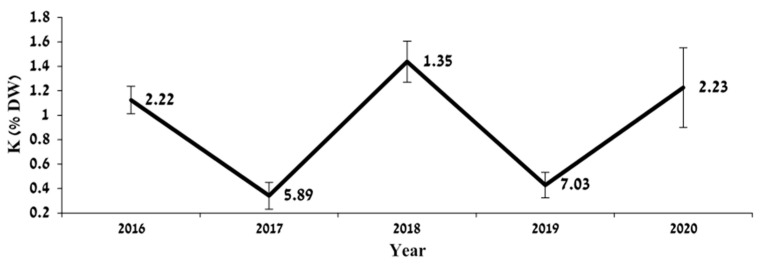
Effect of alternate bearing on K concentration (% dry weight) in leaves sampled in the summer of each experimental year. Values are mean ± SE of five N250-treated plots. Numbers represent each year’s yield (kg/plant).

**Figure 4 plants-12-03132-f004:**
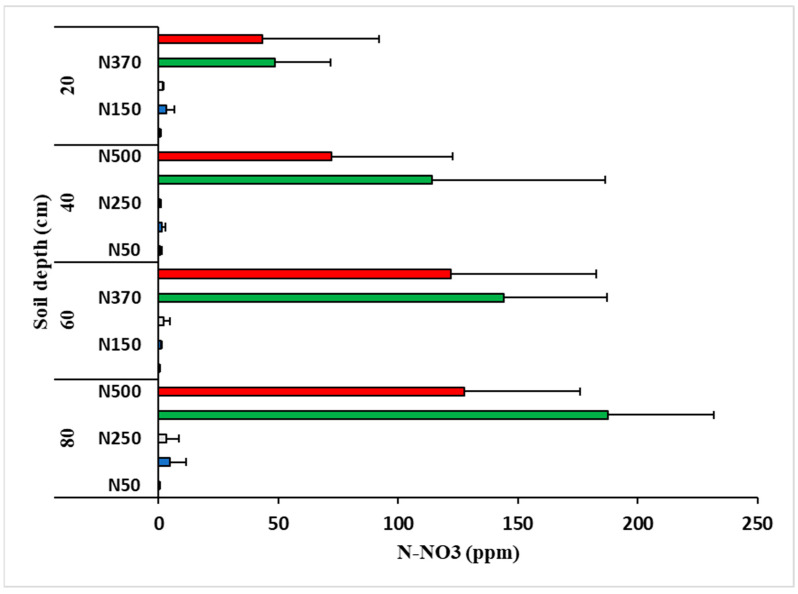
Soil nitrate (N-NO_3_) measured on 14 September 2020 after annual fertilization was completed in all treatments. Values are means of five plots per treatment ± SE.

**Figure 5 plants-12-03132-f005:**
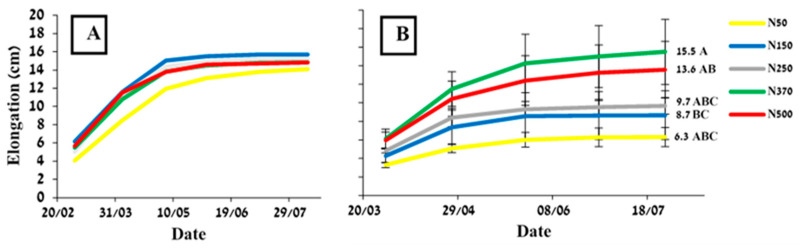
Branch elongation as a function of N-fertilization rate measured in 2020 (**A**), and 2021 (**B**). In 2020, only branches that grew at least 5 cm are included. There were no significant differences between the treatment in 2020 (*p* < 0.05). Values are mean ± SE of five plots per treatment. Numbers are total seasonal elongation (cm). Different letters indicate significant difference in total seasonal elongation (cm) between treatments.

**Figure 6 plants-12-03132-f006:**
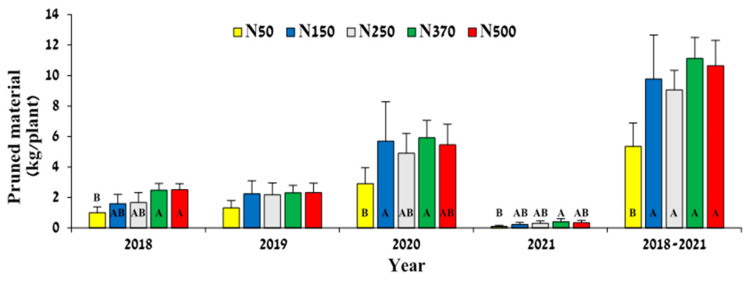
Effect of N-fertilization rate on pruned branch weight, determined after pruning in November of every year. Values are mean ± SE of five plots per treatment. Different letters indicate significant difference (*p* < 0.05) between treatments for each year separately.

**Figure 7 plants-12-03132-f007:**
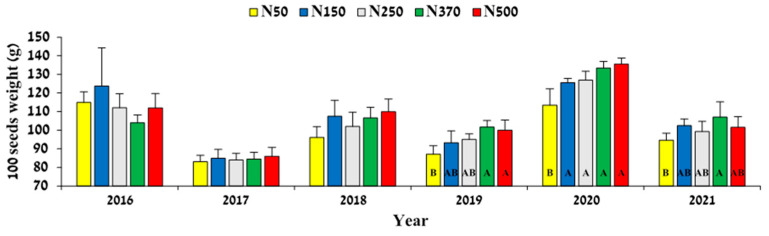
The effect on N-fertilization rate on jojoba seed weight. Values are mean ± SE of five plots per treatment. Different letters indicate significant difference (*p* < 0.05) between treatments in each year.

**Figure 8 plants-12-03132-f008:**
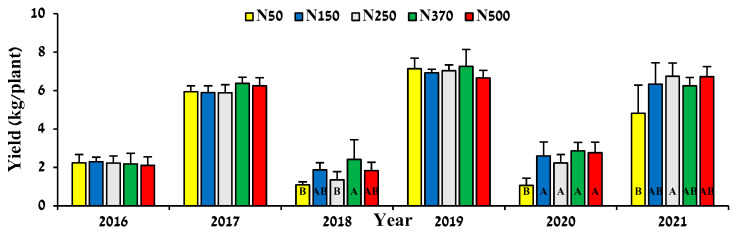
Effect of N fertilization rate on yield. Values are mean + SE of five plots per treatment. Different letters indicate significant difference (*p* < 0.05) between treatments in each year.

**Figure 9 plants-12-03132-f009:**
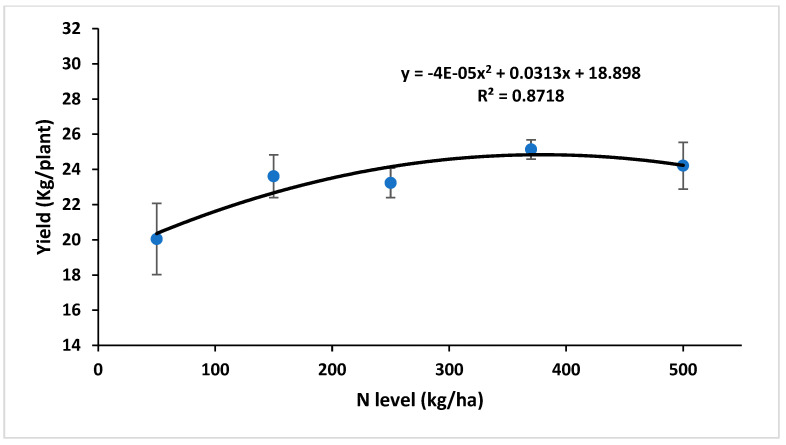
Accumulated yield (2017–2021) as a function of N fertilization rate. Values are mean ± SE of five plots per treatment.

**Table 1 plants-12-03132-t001:** Fertilizer composition of the different N treatments.

Treatment	Composition ^a^	pH	N Total	N-NO_3_	N-NH_4_	P_2_O_5_	K_2_O	Cl	Fe	Mn	Zn	Cu	Mo
			%						ppm				
N50	2-3-9+3	4	2.0	1.0	1.0	3.0	9.0	5.3	300	150	75	11	8
N150	4.5-3-9+3	4	4.5	2.0	2.5	3.0	9.0	6.7	300	150	75	11	8
N250	5-2-7+3	4	5.0	2.5	2.5	2.0	7.0	4.3	300	150	75	11	8
N370	7-2-7+3	4	7.0	3.3	3.7	2.0	7.0	5.3	300	150	75	11	8
N500	10-2.5-5+3	4	10.0	4.7	5.3	2.5	5.0	3.8	300	150	75	11	8

^a^ N-P_2_O_5_-K_2_O.

## Data Availability

Data will be made available on request.

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
