# Peer review of "Optimizing Nitrogen Application for Jojoba under Intensive Cultivation"

_plants, 2023, doi:10.3390/plants12173132_

Round 1

Reviewer 1 Report

The manuscript is will written and adequately describes a well planned study.  The results and conclusions are well described.  References should be consistent, using numbers in brackets (i.e. line 84). Notes to authors should be remover from section 4.3 (lines 420-432).

Author Response

Thank you for the positive feedback on the manuscript.

The reference in line 84 was corrected and the names of the authors and years were replaced by the reference number.

We removed the un-relevant paragraph from section 4.3

Reviewer 2 Report

This manuscript studied the effects of nitrogen fertilizer on jojoba productivity. Different concentrations of nitrogen fertilizer were applied to jojoba cultivation and 6 years survey identified the reasonable condition of nitrogen fertilizer. Although experiments were conducted for many years, novel findings in this manuscript are poor. The contents do not really expand our understanding of nitrogen nutrition in plants.

Author Response

We don't fully agree with the comments on the absence of innovative aspects in our ms. ; defining the nitrogen fertilization requirement for crops is an important topic; over-fertilization with N is one of the leading causes of groundwater contamination and deficiency in N-reduced productivity. This paper is the first one that determines the N requirements for jojoba crops grown in intensive cultivation, and this is defiantly a novel finding. We also reported, for the first time, on the threshold of 1.3% N in diagnostic leaves is the threshold for a deficiency which might be an essential tool for the growers in controlling their N application program. Furthermore, we report that the nitrogen-application rate affects P nutritional status, another innovative aspect which is rarely known from other crops and was reported for the first time in jojoba. An additional novel finding is that Increasing N enhanced seed weight but did not affect wax content.

Reviewer 3 Report

the paper is written well but minor revision need

-the quality of the figure is poor and please add the statistical analysis above the bar not inside

- add statistical analysis of Figure 4

- in figure 7 convert 100 seeds mass to 100 seeds weight

- in figure 8 correct yield hg/plant (delete -1)

- why you don't measure the oil content 

- delete the subtitle in the discussion section

minor corrections are needed

Round 2

Reviewer 2 Report

I think demonstrating jojoba cultivation results  may be new, however, findings from this research lack scientific novelty. N is one of the most important elements in plant nutrition, therefore, numerous studies have been published. Each plant has optimal range of N fertilization, this is not surprising. N fertilization indirectly affects other element accumulation and N fertilization increases seed yield (N is used as seed storage proteins), but not wax (N is not a major component), all these are well-known in plant nutrition. I feel this manuscript contains meaningful results, for example, 6 years cultivation record is valuable, therefore, it's better to publish in suitable journals which mainly focus on agriculture or field crop production.
